# Fusion of Different Image Sources for Improved Monitoring of Agricultural Plots

**DOI:** 10.3390/s22176642

**Published:** 2022-09-02

**Authors:** Enrique Moltó

**Affiliations:** Instituto Valenciano de Investigaciones Agrarias, 46113 Moncada, Spain; molto_enr@gva.es

**Keywords:** image processing, multispectral analysis, Sentinel 2, orthophoto, drone, spectral indices, google earth engine, image similarity

## Abstract

In the Valencian Community, the applications of precision agriculture in multiannual woody crops with high added value (fruit trees, olive trees, almond trees, vineyards, etc.) are of priority interest. In these plots, canopies do not fully cover the soil and the planting frames are incompatible with the Resolution of Sentinel 2. The present work proposes a procedure for the fusion of images with different temporal and spatial resolutions and with different degrees of spectral quality. It uses images from the Sentinel 2 mission (low resolution, high spectral quality, high temporal resolution), orthophotos (high resolution, low temporal resolution) and images obtained with drones (very high spatial resolution, low temporal resolution). The procedure is applied to generate time series of synthetic RGI images (red, green, infrared) with the same high resolution of orthophotos and drone images, in which gray levels are reassigned from the combination of their own RGI bands and the values of the B3, B4 and B8 bands of Sentinel 2. Two practical examples of application are also described. The first shows the NDVI images that can be generated after the process of merging two RGI Sentinel 2 images obtained on two specific dates. It is observed how, after the merging, different NDVI values can be assigned to the soil and vegetation, which allows them to be distinguished (contrary to the original Sentinel 2 images). The second example shows how graphs can be generated to describe the evolution throughout the vegetative cycle of the estimated values of three spectral indices (NDVI, GNDVI, GCI) on a point in the image corresponding to soil and on another assigned to vegetation. The robustness of the proposed algorithm has been validated by using image similarity metrics.

## 1. Introduction

The EU is promoting the use of monitoring systems based on remote sensing for the optimal management of agricultural resources and supporting policies related to climate change with objective data and statistics. The Regional Ministry of Agriculture, Rural Development, Climate Emergency and Ecological Transition of the Generalitat Valenciana (Spain) needs tools to monitor both the impact of human activity and that of agricultural and environmental policy measures on the adaptation and mitigation of climate change at the supralocal level (regional, basin, regional, etc.). Remote sensing monitoring not only has application in the field of large-scale resource management, but it is also important support at the level of individual exploitation through the technologies associated with Precision Agriculture.

The Copernicus programme provides accurate, up-to-date and easily accessible information to improve environmental management, understand and mitigate the effects of climate change and ensure citizen security [1]. The main missions of agricultural interest are already in orbit. Among them, the Sentinel 2 mission stands out, consisting of two identical satellites out of phase 180°, which in Spain provide information every 5 days on the Earth’s surface in 13 visible and infrared spectral bands (VIS-IR) with spatial resolutions between 10 and 60 m [2].

Despite its enormous usefulness in economic and environmental terms, the implementation of technology associated with Precision Agriculture and remote sensing in the Valencian agricultural sector is scarce, due to the lack of information on its possibilities, the nature of production (mainly associated with Mediterranean fruit trees with high added value) and the excess of smallholdings, which collides with the resolution of satellite images. In addition, Precision Agriculture based on remote sensing has been developed mainly in extensive annual crops (corn, wheat, rice, soybeans, rice, etc.). However, in the Valencian Community, its implementation in multi-year crops is of priority interest, such as fruit trees (orange, mandarin, persimmon, pomegranate, avocado, medlar, etc.), olive trees, almond trees and vineyards.

Satellite imagery is an indispensable source of data, but it offers relatively low spatial resolution (currently 3–10 m). In addition, the images produced by satellites equipped with multi or hyperspectral sensors are very susceptible to cloud cover.

Parallel to the rapid evolution of image sources from satellites, new drones are being developed with greater autonomy and versatility that allow obtaining aerial images with multispectral sensors and LIDAR at very high spatial resolution (centimeters). Drones acquire images with a certain temporal flexibility and at a relatively low cost, but they are limited by their flight autonomy, weather conditions and different air safety regulations. A summary of the advantages and disadvantages of both systems is shown in Table 1.

Sentinel 2’s Level 2A products provide data based on constantly calibrated and atmospherically corrected reflectances and are critical for estimating biophysical parameters and detecting land cover. There is a large amount of scientific literature aimed at establishing mathematical relationships between biophysical or soil parameters and certain spectral indices that combine information from two or more bands, such as the Normalized Difference Vegetation Index (NDVI) [3], the Green Normalized Difference Vegetation Index (GNDVI) [4], the Green Chlorophyll Index (GCI) [5], etc. However, the fact that each pixel of the image, at best, covers 10 × 10 m^2^ of soil, makes it impossible to discriminate the value of these indices for soil and vegetation, in plots with crops in which the planting frames are much smaller and the tops do not fully cover the ground, as is the case of the olive grove that concerns us.

One solution to this problem is to combine or merge the higher spatial resolution information provided by aircraft images, with the temporal frequency and image quality of the multispectral sensors installed in the Sentinel 2 missions. Thus, in this article, we use the Sentinel 2 Level 2A product to merge it with orthophotos and images obtained with drones, in order to produce a series of high-resolution synthetic images. The results of the process are of considerable use for applications that require both high spatial resolution and frequent image coverage.

Some satellites incorporate sensors that provide high-resolution panchromatic images and, at the same time, images of different spectral bands at lower resolution. For this reason, since the beginning of remote sensing, image fusion techniques aimed at generating high-resolution multispectral images have been developed. They combine the spectral information of low-resolution data with the high spatial resolution of panchromatic images. Often, these works have been aimed at increasing the contrast or sharpness perceived by a human operator, and therefore, the calibration of spectral radiance or reflectance has not been considered. These approaches work, for example, with images that result from the composition of three bands and their conversion to saturation and tone values in color spaces, such as HSV, the analysis of the main components of the bands [6], or with wavelet transformations [7].

These approximations are not suitable for the problem considered in this work, since one of its goals is to capture quantitative values of changes in surface reflectance, and therefore, in spectral vegetation indexes caused by the phenology of the crop. Attention must be paid both to the effects caused by the different spectral responses of the sensors that are used and to the effects related to the different illumination of the scene. The problem is close to the studies on the elimination of clouds and their shadows in the composition of satellites [8] or aerial images [9]. In them, the effects on histograms are studied in a special way.

Several online platforms are now available to remote sensing users to perform data analysis easily and without increasing the demand for local computing resources. One of these, free to access, is Google Earth Engine [10].

In this context, the major objective of this work is to describe and apply a procedure to generate time series of synthetic RGI images (red, green, infrared) with the same high resolution of orthophotos and drone images, reassigning the gray levels of the latter from the combination of their own RGI bands and bands B3, B4 and B8 of Sentinel 2.

Moreover, the robustness of the proposed methodology is validated by comparing differences between original images and between synthetic ones. Diverse metrics to assess differences between a certain reference image and distorted images have been proposed for image quality assessment. They can be used to quantitatively determine how distinct two images are. Although early methods focused on absolute or relative pixel value differences, these approaches do not consider relevant differences in the structural information present in the images, which is of paramount importance for human visual perception. A comprehensive revision and a proposed metric for structural similarity quantitative assessment (SSIM index) are described in [11]. This metric has been used in this work to measure differences between image pairs as described in Section 2.4.

## 2. Materials and Methods

### 2.1. Image Sources

This application has been developed from the following images, obtained on a plot of olive trees in Villena (Alicante) whose center is located approximately at the coordinates of longitude −0.879577 and latitude 38.596992 (reference system EPSG: 4326):Orthophoto of 2019 of the Valencian Community, obtained from the catalog of Valencian Infrastructure of Spatial Data (https://idev.gva.es/ accessed on 28 July 2022), elaborated at 25 cm/pixel of resolution (Ground Sampling Distance, GSD), from RGBI digital photogrammetric flights made in the period 14 May 2019–30 June 2019 using the camera UltraCamEagle_UC-E-1-50016095-f80 with a Qioptic Vexcel HR Digaron sensor. The Red, Green and Near Infrared (RGI) bands are used in this image. We call this image the original May image.Images were obtained by drone flights made by the company Asdrón Spain using a Parrot Sequoia multispectral camera. The images were processed with the PIX4DMapper package. Of these, the Red, Green and Near Infrared (RGI) bands are used exclusively. The resolution (average GSD) of these images is 6.2 cm/pixel. The flights were made near solar noon, between 13:00 and 14:00 (official local time), on 24 July 2019 (hereinafter referred to as the ‘July’ image), on 14 August 2019 (hereinafter referred to as the ‘original August’ image, used for the validation process) and on 18 September 2019 (hereinafter referred to as the ‘original September’ image). The camera has a sunlight sensor, which performs real-time correction of lighting differences. Reflectance conversion to image grey levels was performed with Pix4D software, which uses previously acquired images of a calibration panel and automatically calculates the appropriate reflectance factor. Image resolution was reduced to 25 cm/pixel by averaging neighboring values and reprojecting.Images of tile 30SXH from the period 1 May 2019–31 October 2019 of the Sentinel 2 satellite constellation, through the MSI sensor (Multi Spectral Instrument) with the processing level 2A. In total, 29 images are obtained with a percentage of clouds lower than 35%. Of each image, only the bands B4, B3 and B8 are considered, whose resolution (GSD) is 10 m/pixel. Each image was subjected to masking both the clouds and their shadows.

All images were reprojected from their original coordinate reference system to EPSG:32630 and clipped to the geometry of the plot. Table 2 summarizes main technical specifications.

### 2.2. Hypotheses

The process of merging the images described in this paper is based on the following hypotheses:a.The spectral quality of the Sentinel 2 data is superior to that of the other sources, similar to how Houborg and McCabeb propose [12] for the sensor carried by Landsat compared to other multispectral sensors.b.Since orthophoto and drone images come from different sensors and were taken on different days, homogenization of reflectance values is required. It is assumed that the histogram matching technique is a sufficient approximation for such homogenization.c.The reflectance values of high-resolution images (orthophoto and drone) must be corrected to be assimilated into the reflectance values provided by Sentinel 2 images.d.The RGI bands of high-resolution images can approximate the B4, B3, and B8 bands of Sentinel 2.e.Reflectance correction is based on the fact that, in each of the bands studied, the average of the pixels of the images of higher spatial resolution that are located below a pixel of Sentinel 2 must coincide with the value of that pixel.f.The correction is linear and depends exclusively on the positions of the pixels in each image.

### 2.3. Description of the Fusion Process

Once the above hypotheses have been accepted, both the histograms of the original image of May (orthophoto) and of the original image of September (drone) are adjusted using the histogram matching technique, with respect to the reference image (‘original image of July’, obtained from drone). From them, the so-called homogenized image of May and homogenized image of September are obtained (the image of July is not transformed since it is the reference).

Subsequently, the fusion process starts. From hypothesis e) we have that in each of the bands we want Equation (1) to be fulfilled:∑ *pf_i_*/*numpix* = *S2_j_*(1)

*pf_i_* being the value to be assigned to the pixel of coordinates *i* in that band, *numpix* the number of pixels of the image of higher resolution that is below the pixel *S2_j_*, of coordinates *j*, in the corresponding band of Sentinel 2.

Hypothesis (f) is formulated as follows, for each band:*pf_i_ = po_i_* × *K_j_*(2)
being *po_i_* the value of the pixel of coordinates *i* in a band of the high-resolution image and *K_j_* the correction factor that must be applied to the high-resolution pixels of that band located under the *S2_j_* pixel.

Combining Equations (1) and (2) yields how to convert the grey level values of each band of the homogenized images into values consistent with the hypothesis (d):*K_j_ = S2_j_ × numpix/∑ po_i_*(3)
*pf_i_ = po_i_ × S2_j_ × numpix/∑ po_i_ = po_i_ × S2_j_/avg_i_*(4)
being *avg_i_* the average of the values of the pixels of the homogenized high-resolution band under the pixel *S2_j_*.

The sentinel2 series of images are then merged with the high resolution homogenized images as follows:a.Sentinel 2 images from the period 1 May 2019–30 June 2019 (11 images) merge with the homogenized image of May, from the orthophoto.b.Sentinel 2 images from the period 1 July 2019–31 August 2019 (12 images) merge with the original Image from Julyc.Sentinel 2 images from the period 1 September 2019–31 October 2019 (6 images) merge with the homogenized image from September.

Later the spectral indices known as NDVI, GNDVI and GCI are calculated on the synthetic images. Figure 1 shows the overall scheme of the procedure.

Parallelly, for validation purposes, Sentinel 2 images from the period 1 August 2019–30 September 2019 (eight images) were merged with the homogenized image of August and the same indexes were calculated. This image series has a time overlap with series c in the period 1 September 2019–30 September 2019 (3 images). Hereinafter, this period is referred as the ‘overlapping period’.

### 2.4. Measurement of Similarity and Validation

As stated in the introduction, the SSIM index has been the chosen metric for measuring similarity between two images (x and y). This index is the combination of three functions that independently compare luminance (*l*(x,y)), contrast (*c*(x,y)) and correlation (*s*(x,y)) of both images:*SSIM*(x,y) = [*l*(x,y)]^α^ · [*c*(x,y)]^β^ · [*s*(x,y)]^γ^(5)

The index was implemented assuming α = β = γ = 1 and coefficients C_1_ = C_2_ = C_3_ = 0 (not shown in Equation (5)), because sums of squared means and sums of squared standard deviations of our images are far from being 0 in our images. As a consequence, we use a particular case of SSIM implementation that corresponds to the Universal Quality Index described in [13]. The sliding window size was set to 11 × 11 pixels for high resolution images and 5 × 5 for Sentinel 2 images. These sizes were selected to keep a ratio image size/window size analogous to the one described in [11] and [13]. Values of *SSIM*, *l*, *c* and *s* were calculated for each band of the compared images

Several similarities have been calculated to assess the robustness of the proposed methodology:a.Similarity between original drone images (July, original August and original September), in order to have an approximate idea of the SSIM initial values.b.Similarity between the reference July image and the homogenized August and September images, in order to assess the effect of the homogenization procedure on similarity.c.Similarity of the Sentinel 2 images during the overlapping period.d.Similarity of the NDVI, GNDVI and GCI images calculated from Sentinel 2 images during the overlapping period.e.Similarity of the NDVI, GNDVI and GCI images obtained from merging the original August and the original September images during the overlapping period.f.Similarity of the NDVI, GNDVI and GCI images obtained from merging the original August or the original September images outside the overlapping period.

### 2.5. Algorithms’ Implementation

The algorithms were developed using the JavasScript API and the code editor of the Google Earth Engine platform. High-resolution images were imported as assets to the platform. Likewise, Figure 2 and Figure 3 have been generated using these tools.

## 3. Results

### 3.1. Homogenization of High-Resolution Images

Figure 2 shows the results of the homogenization of the infrared bands of the May and September images with respect to the July image. It is observed that the original histograms (a), (b) and (d) have different forms. After this operation, the distribution of the histograms of the homogenized images (c) and (e) is similar to that of reference (a). The procedure is applied similarly to the red (R) and green (G) bands.

### 3.2. Merging Images from Sentinel 2 Images with High Resolution Images. NVDI Estimation

Figure 3 shows the first example of the application of this work. On the left (Figure 3a,c) are two NDVI images calculated from Sentinel 2 images (processing level 2A) obtained on 25 July and 18 September On the right (Figure 3b,d) shows the result of merging these images with the corresponding images from a drone (July image and Homogenized September image). In Figure 3a,c, due to the lower spatial resolution of the Sentinel 2 images, NDVI values can only be observed corresponding to large areas of the plot and, as expected, without being able to distinguish between soil and vegetation.

After the fusion procedure, the NDVI values are reassigned in the high-resolution images, and therefore, in the (b) and (d) images, both soil and vegetation are clearly distinguished.

### 3.3. Series of Values of the Spectral Indices Estimated from the Synthetic Images of High Resolution

In this section, we show another example of the application of this work. The remapping of pixel values using the blending algorithm applies to the entire Sentinel 2 image series, as indicated in Section 2.3. In this way, a temporal sequence of fused images can be created that allows to simulate the evolution of the spectral indices in these synthetic images.

Figure 4 shows the simulation of the time evolution of the NVDI, and GNDVI indices that are generated from the high-resolution merged images (dashed lines) and from the original Sentinel 2 images (continuous lines). Figure 5 shows the simulation of the temporal evolution of the GCI index.

In both figures, the green lines of stripes (ndvi-veg, gndvi-veg and gci-veg) represent the values obtained from a point in the merged images (high resolution) associated with vegetation. The brown dotted lines (ndvi-soil gndvi-soil and gci-soil) represent the values obtained from a point in the fused (high resolution) images associated with soil. The continuous gray lines ndvi-S2, gndvi-S2 and gci-S2 correspond to the homologous pixel in the Sentinel 2 images.

It is verified that after fusion, the indices are higher at the point that corresponds to the vegetation and lower at the point that corresponds to the ground, while the area that covers a pixel of the original Sentinel 2 images (10 * 10 m in the bands considered) maintains intermediate values.

### 3.4. Validation

#### 3.4.1. Similarity between Original Drone Images

Table 3 displays the similarity values of the RGI images obtained by drone in August and September with respect to the July image. SSIM ranged from 0.543 to 0.842. The major dissimilarities (lower values) were found in the correlation parameter (*s*) in almost all bands. In particular, the Near Infrared band (I) had the lowest similarity values on both dates. Identical data behavior was observed when comparing August and September images (data not shown), with SSIM values of 0.757, 0.805 and 0.550, respectively, for the RGI bands.

#### 3.4.2. Effect of the Homogenization Procedure

Table 4 expresses the similarity values of the homogenized August and September images with respect to the July image. An increase in *SSIM* was clearly observed (values ranged from 0.584 to 0.892), thus favoring the hypothesis that the procedure was able to slightly dilute some differences between acquisition dates. The increase in SSIM was due to the increase in the similarities in contrast (*c*), correlation (*s*) and luminance (*l*) in all bands. As before, the Near Infrared band (I) had the lowest similarity values on both dates. Analogous values and behavior were observed when comparing August and September homogenized images (data not shown), with *SSIM* values of 0.846, 0.837 and 0.615, respectively, for the RGI bands.

#### 3.4.3. Similarity of the Sentinel 2 Images during the Overlapping Period

Table 5 displays the similarity values of the RGI images obtained from Sentinel 2 during the overlapping period (September 2019). In this month, three images were used, acquired on the 3rd, 18th and 28th. Data presented in this table utilize the image obtained on September 3 as a reference. *SSIM* values are higher than in the previous cases, ranging from 0.964 to 0.983. Values of all parameters are higher than 0.900 in all bands. Analogous values were obtained when comparing September 18 and September 28 images, with *SSIM* of 0.989, 0.986 and 0.995, respectively, for the RGI bands.

#### 3.4.4. Similarity of NDVI, GNDVI and GCI Images Calculated from Sentinel 2 Images during the Overlapping Period

NDVI, GNDVI and GCI images calculated from the Sentinel 2 series during the overlapping period also revealed a great level of similarity between them (data not shown), SSIM ranging from 0.986 to 0.998.

#### 3.4.5. Similarity of NDVI, GNDVI and GCI Images Obtained from Merging the Original August and Original September Images during the Overlapping Period

Table 5 displays crucial data for testing the validity of the whole procedure described in this paper. It compares two sets of resulting NDVI, GNDVI and GCI images: those obtained by merging the August image and those obtained by merging the September image during the three days of the overlapping period. Consequently, it provides information about how sensible the procedure is to changes imputable to differences between the original drone images. It can be observed that the resulting NDVI, GNDVI and GCI images obtained from the two original drone images had strong similarities, with SSIM ranging from 0.876 to 0.894. Major dissimilarities were caused by lower values of the correlation parameter (s).

These data suggest that this effect is linked to the dissimilarities of the original drone images with respect to this parameter before (Table 3) and after homogenization (Table 4).

On the other hand, it must be highlighted that the similarity values of NDVI, GNDVI and GCI images obtained from the merged images with different sources (Table 6) are lower than those obtained from comparing analogous images from Sentinel 2 data on different days in the overlapping period (Section 3.4.4). This can be attributed to two factors: (a) Sentinel images have coarser resolution and (b) changes of these spectral indexes are relatively slow.

#### 3.4.6. Similarity of the NDVI, GNDVI and GCI Images Obtained from Merging the Original August or the Original September Images Outside the Overlapping Period

In this section, we provide data on the similarity of NDVI, GNDVI and GCI images calculated on two different days outside the overlapping period. It must be recalled that the series of synthetic images obtained by merging the August image covered the period 1 August 2019–30 September 2019; therefore, they did not overlap during August with the series of synthetic images obtained by merging the September image (1 September 2019–31 October 2019). Likewise, this last one did not overlap the previous one in October. 

Comparing two NDVI, GNDVI and GCI images obtained on different days of August values of *SSIM* were, respectively, 0.993, 0.993 and 0.985. In the same way, in October, values of *SSIM* were, respectively, 0.995, 0.994 and 0.991, very close to those obtained directly from Sentinel 2 images during the overlapping period (Section 3.4.4).

## 4. Discussion

This article describes an RGI image fusion procedure in which information from several sources with different spatial and temporal resolutions are originally merged, using a simple, easy-to-implement novel algorithm that can be applied under the assumptions expressed in Section 2.2. The assumption of higher spectral quality of Sentinel 2 data is adopted by Houborg and McCabeb [12], their and other works use two satellite sources. In these cases, the temporal resolution of higher resolution images is equal to or better than those of Sentinel 2. They use Sentinel 2 basically to correct image series obtained from higher spatial resolution satellites. However, this work explores a different application in the temporal dimension, which is the possibility of using low temporal resolution information to correct Sentinel 2 temporal series, thus providing a practical means to increase the applications of this satellite imagery.

The paper provides two examples of agricultural applications of the proposed algorithm. The proposed algorithm re-distributes the low-resolution pixel values of Sentinel 2 in higher resolution pixels and provides a robust method to assess the evolution of spectral indices of points of the soil and points of the vegetation that cannot be distinguished in the original Sentinel 2 images. Therefore, it has demonstrated that it can be applied in a series of Sentinel 2 images, where pixel size is much larger than trees. 

This same algorithm can obviously be used in images composed of more bands, which expands the number of applications. For instance, in agronomy, it can be used for the determination of spectral indices that also use the blue band, such as Enhanced Vegetation Index (EVI) [14], Green Leaf Index (GLI) [15], Triangular Greenness Index (TGI) [16], Normalized Pigment Chlorophyll Ratio Index (NPCRI) [17], or Structure Insensitive Pigment Index (SIPI) [18]. However, since no information on short-wave infrared (SWIR) is provided by the cameras employed to create the orthophotos or mounted in the drones used in this work, the algorithm is not intended to be directly applied to calculate indexes related to water or moisture contents in the soil or vegetation, such as the Moisture Stress Index (MSI) [19] or the Normalized Difference Water Index (NDWI) [20]. SWIR information from RGI can be estimated using different methods, such as those for pansharpening (for instance by linear regression or random forest-based procedures), then the proposed algorithm could be applied. Nevertheless, this has been considered out of the scope of the present work, since it is devoted to demonstrating a new fusion method.

Although this work represents a significant advance in remote sensing using satellite images, it should be borne in mind that the spectral response of the sensors that acquired the images used in this work is not exactly coincidental, so an approximation has been made. This approximation can be improved by using calibration transfer techniques, commonly used in spectrophotometry, such as the ones described by Alamar et al. [21].

On the other hand, it is necessary to recognize that other methods for homogenizing the spectral responses between different sensors can be found in the literature. Histogram matching has been used in this work due to its simple implementation. Methods, such as those related to the reproduction of colors in RGB images, which use transformations to other color spaces, such as HSI or lαβ could be also adapted to this problem. In this regard, it is recommended to consult the work of Reinhart et al. [22].

It is also important to note that the results of the algorithm presented here depend on the reference image used in the homogenization process, which requires optimization of this selection. This is especially important when more high-resolution images are available.

It must be also emphasized that the work is based on the hypotheses described in Section 2.2 and there are alternatives to hypotheses (e) and (f) that could improve our results. Instead of using linear functions, such as the inverse average function to re-distribute the Sentinel 2 pixel values, other functions, including non-linear ones, could be employed. They have not been addressed in this article for the sake of the simplicity of their handling.

This work also proposes dissimilarity measurements employed for assessing image distortion as objective, quantitative means for estimating distances between images. Such tools served to demonstrate the robustness of the proposed methodology.

## 5. Conclusions

This article describes a simple method for fusing images of high spectral quality, low spatial resolution, and high temporal resolution, from the Sentinel 2 mission, with images of lower spectral quality, low temporal resolution and higher spatial resolution, such as those from orthophotogrammetry and drone flights.

After homogenization of the observed reflectances, synthetic images can be generated from the time series of the Sentinel 2 mission, in which the values of the gray levels of the higher resolution images are reassigned in a differentiated way can be obtained. In addition, the evolution of various spectral indices on the ground and on vegetation can be estimated in a coherent way.

Although this work shows an application example of the proposed fusion algorithm, it can be further generalized to allow further exploitation of Sentinel 2 data to characterize the agronomic conditions of plots producing woody annual crops.

Finally, the robustness of the procedure has been validated by using different image similarity measurements to analyze the effects of employing different drone images and to compare against the similarity of images in the Sentinel 2 series.

## Figures and Tables

**Figure 1 sensors-22-06642-f001:**
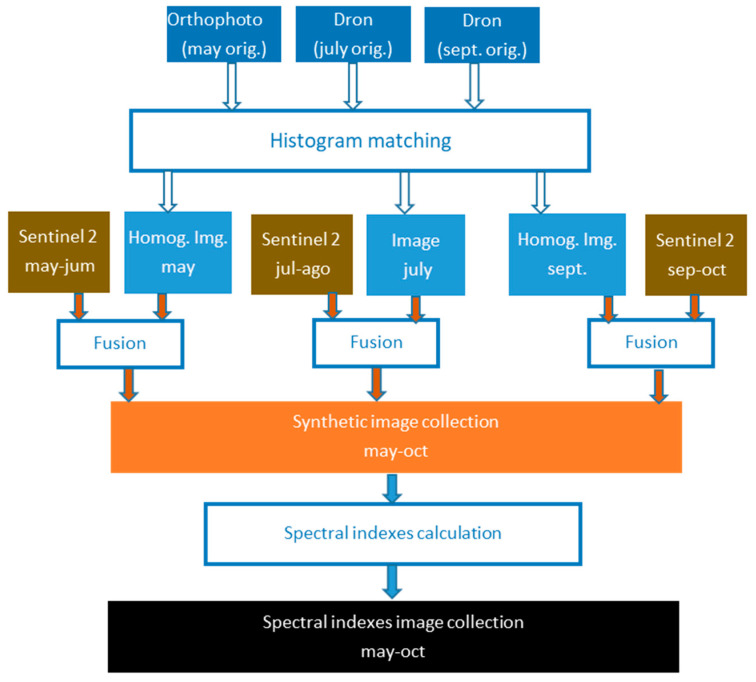
General outline of the process for the fusion of images used in this work.

**Figure 2 sensors-22-06642-f002:**
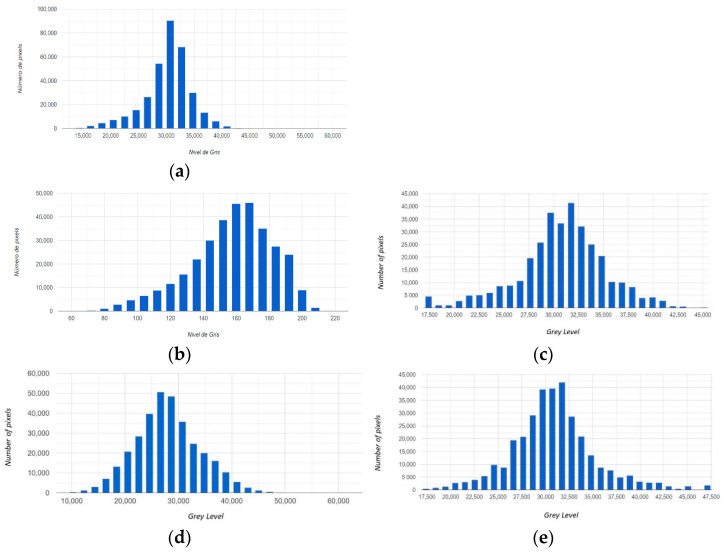
Histogram matching. (**a**) Histograms of band I (Infrared) of the original Image of July (reference). Following images represent histograms before (left) and after correction: (**b**) the original image of May, (**c**) the homogenized image of May, (**d**) the original image of September, (**e**) the homogenized image of September.

**Figure 3 sensors-22-06642-f003:**
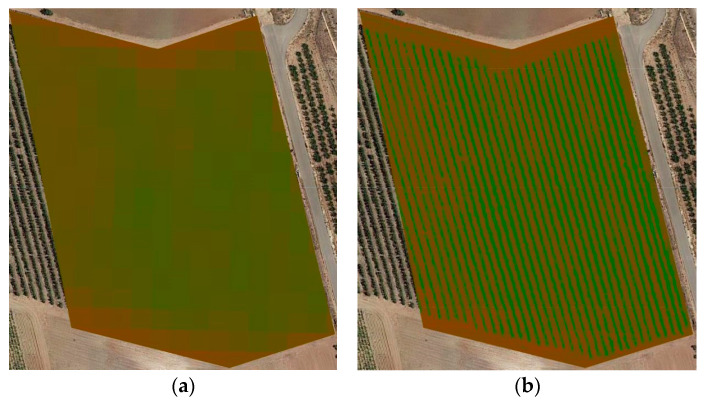
(**a**) NDVI image generated from the Sentinel 2-2A image of 25 July 2019, (**b**) Corresponding merged NDVI image (**c**). -NDVI image generated from the Sentinel 2-2A image of 18 September 2019, (**d**) Corresponding merged NDVI image. Low NVDI values are represented in reddish colors, high values-in greenish colors.

**Figure 4 sensors-22-06642-f004:**
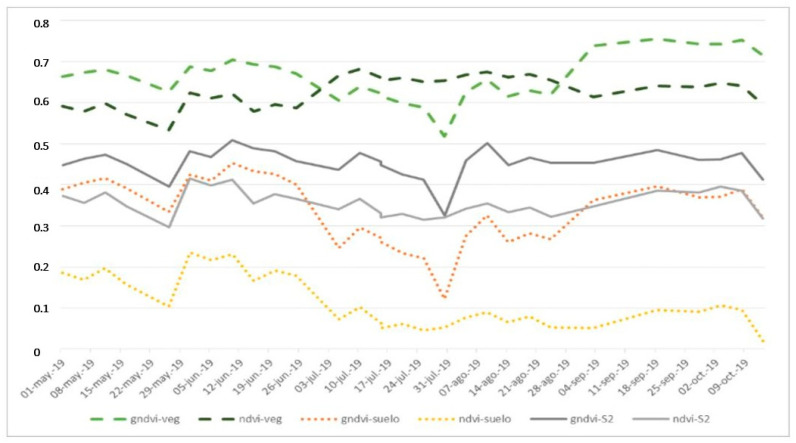
Evolution of the NVDI and GNDVI spectral indices of a point associated with vegetation (green striped lines: ndvi-veg, gndvi-veg), of a point associated with soil (brown dotted lines: ndvi-soil gndvi-ground) and of the pixel in which these points are located in the Sentinel 2 images (gray continuous lines: ndvi-S2, gndvi-S2). Period: 1 May to 31 October 2019.

**Figure 5 sensors-22-06642-f005:**
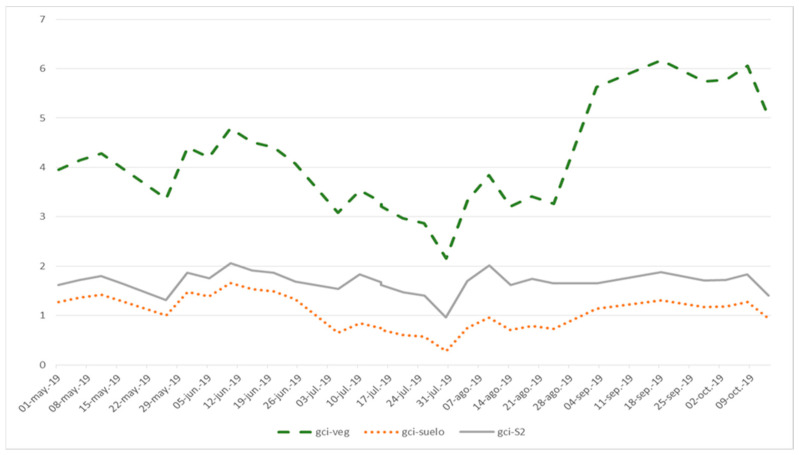
Evolution of the GCI spectral indices of a point associated with vegetation (green striped line: gci-veg), of a point associated with soil (brown dotted line: gci-soil) and of the pixel in which these points are located in the Sentinel 2 images (gray continuous line: gci-S2). Period: 1 May to 31 October 2019.

**Table 1 sensors-22-06642-t001:** Summary of the agricultural use potentialities of remote sensing using satellites and drones equipped with multi and hyperspectral sensors or LIDAR.

	Satellites		Aerial Vehicles	
Information	Resolution	Domain	Resolution	Domain
Temporal	weekly/daily	Past-Future	Programmable	Present-Future
Spatial	1–60 m	0.5 ha-worldwide	0.01 m–0.6 m	0.5-tens of ha
Radiometric	Multi-hyperspectral	Visible microwave	Multispectral	VIS-NIR-LIDAR
Cost	Free/Paid		Paid	
	**Major limitations for agricultural use**			
	Atmospheric conditions		Autonomy	
	Spatial resolution		Air regulations	
	Development costs		Atmospheric conditions	
			Costs	
	**Major advantages for agricultural use**			
	Amount of information		User-oriented image acquisition	
	Images and tools for open use		Flexibility	
	Time series			

**Table 2 sensors-22-06642-t002:** Summary of main technical specifications of images and bands used in this work.

	Orthophoto	Drone Images	Sentinel 2 Images
Camera	UltraCamEagle_UC-E-1-50016095-f80. Qioptic Vexcel HR Digaron sensor	Parrot Sequoia multispectral	MultiSpectral Instrument (MSI)
Bands *	Near Infrared, Red, Green, BlueI	Near Infrared, Red Edge, Red, Green	13 Bands Blue-SWIR B8 (Near Infrared), B4 (Red), B3 (Green)
Original Nom. Resolution	0.25 m	0.062 m	10 m
Resolution employed	0.25 m	0.25 m	10 m
Acquisition dates	14 May 2019–30 September 2019 (precise date unknown)	24 July 2019 (‘July’) 14 August 2019 (‘August’, validation) 18 September 2019 (‘September’)	1 May 2019–31 October 2019 (29 images, separated 5–10 days, depending on cloud coverage)

* Available bands. Only underlined bands are used in this work.

**Table 3 sensors-22-06642-t003:** The similarity between original drone images. Reference: July image.

	August Image			September Image		
Parameter/Band	R	G	I	R	G	I
*c*	0.936	0.938	0.972	0.890	0.923	0.918
*s*	0.935	0.928	0.743	0.895	0.838	0.597
*l*	0.962	0.961	0.996	0.935	0.947	0.991
*SSIM*	0.842	0.837	0.719	0.744	0.732	0.543

**Table 4 sensors-22-06642-t004:** Similarity between homogenized drone images. Reference: July image.

	August Image			September Image		
Parameter/Band	R	G	I	R	G	I
*c*	0.953	0.958	0.969	0.992	0.987	0.963
*s*	0.916	0.917	0.742	0.901	0.841	0.608
*l*	0.977	0.986	0.996	0.998	0.998	0.998
*SSIM*	0.852	0.866	0.717	0.892	0.829	0.584

**Table 5 sensors-22-06642-t005:** Similarity of the Sentinel 2 images during the overlapping period. Reference: September 3 image.

	Sentinel 2–September 18			Sentinel 2–September 28		
Parameter	R	G	I	R	G	I
*c*	0.988	0.992	0.996	0.988	0.995	0.994
*s*	0.991	0.989	0.991	0.994	0.995	0.989
*l*	0.986	0.990	0.996	0.984	0.992	0.994
*SSIM*	0.964	0.972	0.983	0.967	0.982	0.977

**Table 6 sensors-22-06642-t006:** Similarity of NDVI, GNDVI and GCI images obtained from merging the original August and original September images during the overlapping period.

	September 3			September 18			September 28		
Parameter	NDVI	GNDVI	GCI	NDVI	GNDVI	GCI	NDVI	GNDVI	GCI
*c*	0.981	0.980	0.967	0.981	0.980	0.967	0.982	0.980	0.967
*s*	0.921	0.914	0.918	0.921	0.914	0.918	0.921	0.914	0.918
*l*	0.970	0.998	0.992	0.973	0.998	0.992	0.975	0.998	0.992
*SSIM*	0.876	0.894	0.880	0.879	0.894	0.880	0.881	0.894	0.880

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
