# Peer review of "Fusion of Different Image Sources for Improved Monitoring of Agricultural Plots"

_sensors, 2022, doi:10.3390/s22176642_

Round 1
Reviewer 1 Report
This paper provides a remote sensing data fusion method that combines Sentinel 2 image hyperspectral, high temporal resolution and high spatial resolution images obtained by UAVs, which can extract more accurate spectral indices. This method may have certain application prospects.
There are still some issues that need to be clarified.
1. If possible, please use a table showing the technical parameters and acquisition times of the orthophoto, UAV, and Sentinel 2 images used in this article.
2. It is necessary to introduce the principle and steps of the radiation correction method of UAV images in detail.
3. The three indices selected in the verification are all related to vegetation, whether it is possible to add a non-vegetation index, such as soil moisture, to comprehensively verify the reliability of this fusion technology.
4. Comparing with other hyperspectral and high spatial resolution image fusion techniques, the innovation of the method proposed in this paper needs to be explained in detail in the Discussion section.
5. The wavy line below the text in Figure 1 should be removed.
6. The text should be more clearly in Figure 2.
Author Response
- If possible, please use a table showing the technical parameters and acquisition times of the orthophoto, UAV, and Sentinel 2 images used in this article.
A new table (Table 2) has been included with a summary of the technical information of three image sources.
2. It is necessary to introduce the principle and steps of the radiation correction method of UAV images in detail.
Information about reflectance calibration of UAV images has been incorporated in section 2.1.
The grey level correction applied in this work is the same for both drone and orthophoto bands of each image (histogram matching with respect of the drone ‘July’ image and application of equation 4). Redaction of section 2.3 has been modified to make this fact clearer.
3. The three indices selected in the verification are all related to vegetation, whether it is possible to add a non-vegetation index, such as soil moisture, to comprehensively verify the reliability of this fusion technology.
Soil moisture indexes have not been included in this article since they require infrared bands that are not provided neither by the camera used for the orthophotos nor the one mounted on the drones. However, this issue has been incorporated in the second pargraph the discussion section.
4. Comparing with other hyperspectral and high spatial resolution image fusion techniques, the innovation of the method proposed in this paper needs to be explained in detail in the Discussion section.
Major innovative aspects of this work have been included in the first paragraph discussion section as requested by the reviewer.
5. The wavy line below the text in Figure 1 should be removed.
Done
- The text should be more clearly in Figure 2.
Text in the 5 figures has been improved.
Reviewer 2 Report
Thank you for providing me with the opportunity to read “Fusion of different image sources for improved monitoring of agricultural plots”. I have the following comments:
1) The contributions of this research should be clearly highlighted.
2) The multi-sources remote sensing images have been used by multiple studies for agricultural plots monitoring. Therefore, the authors need to clearly discuss the key innovations of their studies compared to these recently published papers.
3) In table 1, the “Atmospheric conditions” is repeated.
4) Figure 1 and 2d needs to be re-edited.
5) The experiment in this paper is weak, and it is difficult to effectively verify the improvement effect of agricultural plot monitoring
6) The lack of comparative experiments cannot verify the advantages of the proposed method over the traditional method.
Author Response
1) The contributions of this research should be clearly highlighted.
This has been addressed in the discussion section.
2) The multi-sources remote sensing images have been used by multiple studies for agricultural plots monitoring. Therefore, the authors need to clearly discuss the key innovations of their studies compared to these recently published papers.
This has been addressed in the discussion section
3) In table 1, the Atmospheric conditions is repeated.
This is because atmospheric conditions affect both satellite and drone image acquisition.
4) Figure 1 and 2d needs to be re-edited.
They have been improved.
5) The experiment in this paper is weak, and it is difficult to effectively verify the improvement effect of agricultural plot monitoring.
The paper basically presents a simple, easy to implement algorithm for merging information from image sources with different spatial and temporal resolutions, which is its major achievement. It has also demonstrated that it can be applied for the calculation of vegetation indexes of soil and vegetation in series of Sentinel 2 images, where pixel size is much larger than trees, which has important applications for agricultural plot monitoring. These ideas are now highlighted in the discussion section.
6) The lack of comparative experiments cannot verify the advantages of the proposed method over the traditional method.
There is no traditional method for image fusion. However, this issue has been addressed in the discussion section.
Round 2
Reviewer 2 Report
Revision has been done. so this is accepted for publications